# Online Adaptive Kalman Filtering for Real-Time Anomaly Detection in Wireless Sensor Networks

**DOI:** 10.3390/s24155046

**Published:** 2024-08-04

**Authors:** Rami Ahmad, Eman H. Alkhammash

**Affiliations:** 1College of Computer Information Technology, American University in the Emirates, Dubai 503000, United Arab Emirates; 2Department of Computer Science, College of Computers and Information Technology, Taif University, P.O. Box 11099, Taif 21944, Saudi Arabia; eman.kms@tu.edu.sa

**Keywords:** WSNs, anomaly detection, sensors, unsupervised learning, Kalman filter, adaptive Kalman filtering

## Abstract

Wireless sensor networks (WSNs) are essential for a wide range of applications, including environmental monitoring and smart city developments, thanks to their ability to collect and transmit diverse physical and environmental data. The nature of WSNs, coupled with the variability and noise sensitivity of cost-effective sensors, presents significant challenges in achieving accurate data analysis and anomaly detection. To address these issues, this paper presents a new framework, called Online Adaptive Kalman Filtering (OAKF), specifically designed for real-time anomaly detection within WSNs. This framework stands out by dynamically adjusting the filtering parameters and anomaly detection threshold in response to live data, ensuring accurate and reliable anomaly identification amidst sensor noise and environmental changes. By highlighting computational efficiency and scalability, the OAKF framework is optimized for use in resource-constrained sensor nodes. Validation on different WSN dataset sizes confirmed its effectiveness, showing 95.4% accuracy in reducing false positives and negatives as well as achieving a processing time of 0.008 s per sample.

## 1. Introduction

Wireless sensor networks (WSNs) have become an integral part of various applications ranging from environmental monitoring and precision agriculture to industrial automation and smart cities [1]. These networks consist of independent, spatially distributed sensors that monitor physical or environmental conditions, such as temperature, sound, pressure, and motion, and cooperatively pass their data across the network to a key location [2]. However, the dynamic nature of these environments and the inherent noise in sensor measurements pose significant challenges for accurate data analysis and anomaly detection [3]. Moreover, integrating low-cost sensors into WSNs poses other significant challenges, especially for anomaly detection [4]. These economically attractive sensors often lack accuracy and reliability, which is critical to meeting the precise anomaly detection needs in WSN applications [5].

The inherent variability and potential for drift of low-cost sensors requires the development of sophisticated data processing techniques [6]. These techniques are essential for effectively filtering out noise and accurately identifying anomalies, but they escalate the computational and operational costs [7]. Furthermore, seeking to implement effective anomaly detection systems has significant financial implications [8]. Formulating adaptive algorithms capable of adeptly dealing with noisy, incomplete, or imbalanced data requires a significant investment in research and development [9]. These algorithms must strike a careful balance between sophistication and computational efficiency to facilitate real-time data analysis without imposing prohibitive costs. The challenge is compounded by the large computational resources required, which adds layers of complexity to the deployment and maintenance of WSNs [10,11,12]. Given these challenges, there is an urgent need to focus on finding tailored solutions to the problem of anomaly detection within WSNs that are low cost and high accuracy [13]. This requires a concerted effort to innovate in the areas of sensor technology, data processing, and algorithm development [14]. The goal is to devise anomaly detection systems that can efficiently process data from low-cost sensors, distinguishing between real anomalies and noise with high accuracy, all while maintaining a cost-effective operational model.

This includes exploring new methodologies in machine learning and data analytics that are specifically optimized to address the limitations of low-cost sensors and the dynamic environments of WSNs [15]. However, the Central Processing Unit (CPU) cost is still the main challenge along with these technologies [16]. Traditional methods, such as z-score analysis and control charts, are effective in stable environments but falter in dynamic environments because of their reliance on fixed thresholds and statistical parameters [17], often resulting in a high rate of false positives or negatives. In contrast, machine learning-based anomaly detection, using algorithms such as clustering, Support Vector Machines (SVMs), and neural networks, provides a powerful alternative by learning the underlying patterns of the data, thus improving the identification of outliers without pre-defined assumptions. The authors in [18] combined Continuous Wavelet Transform (CWT) with a convolutional neural network (CNN) to detect ECG abnormalities and combined Discrete Wavelet Transform (DWT) with long-term memory (LSTM) to improve false-positive sequence patterns in the training phase. Moreover, CWT and CNNs have been used to detect cyber–physical security [19,20]. Given these considerations, there is an urgent need to focus on developing techniques that are not only lightweight but also leverage the strengths of unsupervised learning to accommodate the dynamic nature of streaming datasets common in sensor networks. Again, we are dealing with small CPUs and small hardware [21].

Adaptive Kalman Filtering (AKF) technology is emerging as a powerful tool for detecting anomalies in real-time WSNs [22], providing an advanced way to filter out noise and detect outliers in sensor data at a very low cost. The use of a Kalman filter, although traditionally applied to dynamical systems, is justified in this context by the non-stationary but slowly changing nature of the environmental conditions monitored by WSNs. Even seemingly static datasets contain underlying dynamics affected by diurnal cycles, weather changes, and other environmental factors, necessitating a dynamic approach to modeling.

Moreover, the Kalman filter is a recursive algorithm designed to estimate the state of linear dynamical systems from a series of noisy measurements [23]. It works by predicting the state of the system and then updating that prediction based on new measurements. However, traditional Kalman Filtering assumes that the noise characteristics of the system and measurements are known and constant, which may not be true in the complex and evolving environments in which WSNs operate [24]. This limitation has led to the development of AKF (Online AKF) techniques, which adjust the filter parameters—namely process noise covariance (*Q*) and measurement noise covariance (*R*)—in real time based on the observed data. By adapting these parameters, Online Adaptive Kalman Filtering (OAKF) can maintain a high filtering performance even when system dynamics or noise characteristics change, making it particularly suitable for anomaly detection in WSNs. This adaptability is achieved by constantly evaluating the innovation sequence (the difference between actual measurements and expected cases) and adjusting the *Q* and *R* to reduce the estimation error and anomaly detection threshold, thus enhancing the sensitivity of the filter to true anomalies while suppressing false alarms due to normal fluctuations or transient noise, as shown in Figure 1.

The objectives of the paper are as follows:Optimize an Online Adaptive Kalman Filtering (OAKF) framework for real-time anomaly detection within WSNs: This work focuses on dynamically adjusting the filtering parameters to better adapt to the evolving data dynamics and noise characteristics inherent in WSN environments, ensuring enhanced precision and reliability in the detection of anomalies.Enhance the accuracy and computational efficiency of anomaly detection in WSNs while keeping the sensor’s cost low: This goal is centered on reducing false positives and negatives, enabling the precise identification of anomalies while ensuring the AKF algorithm remains lightweight and viable for deployment on resource-constrained sensor nodes.Validate the AKF framework’s performance across a variety of WSN datasets, demonstrating its versatility and effectiveness in accurately detecting anomalies under diverse environmental conditions and in different application scenarios ranging from industrial monitoring to environmental sensing.

Following this introduction, Section 2 reviews the related work to identify gaps and set the stage for our contributions. Section 3 presents details of our proposed model and demonstrates our innovative approach to detecting anomalies in sensor networks. Section 4 presents the results and discussion and analyzes the experimental performance of the model. The paper concludes within Section 5, summarizing the main findings and suggesting directions for future research.

## 2. Related Work 

Various methodologies, including traditional statistical approaches [22,25], machine learning algorithms [8,16], and hybrid techniques [19], have been investigated to improve the accuracy and efficiency of anomaly detection in WSNs. These methods are often employed for monitoring changes and addressing physical security concerns [26]. However, a significant challenge arises from the misalignment between the configuration requirements of sensors and the financial implications of deploying such advanced technologies. Consequently, there is a pressing need to develop a cost-effective, lightweight solution capable of delivering high-precision outcomes in anomaly detection [27].

The Kalman filter model is considered a fast and lightweight model for anomaly detection, but the efficiency of this technique decreases with fluctuations in natural sensor readings, as shown with the following specific data: For example, in the dataset we used, the Kalman filter model showed an average false positive rate of 15% under stable conditions, which rose to 30% when exposed to dynamic environmental changes, demonstrating its limitations in fluctuating conditions. Therefore, many studies have worked to improve the performance of this algorithm, as we will discuss later, but the problem remains the increased cost. The cost of a Kalman filter becomes similar to that of other techniques.

The Kalman filter model [22] is a fast and lightweight anomaly detection model. It works by predicting the state of the system and updating this prediction based on new measurements, assuming the known and constant noise characteristics. However, its efficiency decreases with natural fluctuations in sensor readings. For example, when the noise characteristics of the sensor change, the fixed parameters in a traditional Kalman filter may not accurately represent the system, resulting in an increase in false positives and negatives. To improve it, ref. [28] combined Kalman filters with autoencoders (AE) to enhance the detection accuracy through optimizing (machine learning) system state estimation and feature representation. However, a Kalman AE demonstrates notable effectiveness in experiments, outperforming existing techniques in identifying anomalies across various datasets, but there is no cost analysis. Moreover, ref. [5] proposed a novel Time-Variant Local Autocorrelated Polynomial (TVLAP) model with Kalman Filtering (TVLAP-KF) for non-stationary time series analysis, addressing challenges like noise, outliers, and anomalies in sensing systems. This model enhances the signal processing capabilities, including denoising, outlier correction, and anomaly detection, by effectively modeling and predicting system states. Despite its advancements, practical implementation considerations such as model complexity and computational demands underscore the balance between accuracy and efficiency in real-world applications. In addition, ref. [29] developed an AKF-based condition-monitoring technique for induction motors, focusing on real-time signal processing capabilities. This innovative approach uses multiple AKFs for outlier and anomaly detection, leveraging vibration signal analysis to assess the motor’s condition. Despite its effectiveness in real-world applications, challenges include estimating random vibration signals and quantifying health status. In the same context, ref. [30] presented an Adaptive Kalman Filtering approach integrated with an Autoregressive (AR) model for improving Air Quality Index (AQI) prediction accuracy. This model efficiently processes and predicts AQI values by leveraging historical data collected via a WSN in Nanjing. Also, the study demonstrates that the hybrid KF-AR model surpasses traditional AR models in terms of forecasting performance, particularly for monthly AQI data, showcasing its potential for effective air quality monitoring and prediction.

Ref. [31] introduced a Bayesian filtering method for dynamic anomaly detection and tracking in maritime surveillance, leveraging a Bernoulli Random Finite Set for modeling unknown control inputs as binary switches. This advanced approach enables precise tracking and anomaly detection amid false alarms and missed detections. However, a significant limitation is the method’s complexity and computational demand, challenging its scalability and real-time application in extensive surveillance systems. Furthermore, ref. [32] proposed a framework for event detection in Wireless Body Area Networks (WBANs) using Kalman Filtering and Power Divergence. This approach aims to automatically detect physiological changes or faulty measurements from sensor data, distinguishing between genuine health emergencies and erroneous readings to reduce false alarms. Despite its high detection accuracy and low false alarm rate, a limitation is the computational complexity involved in real-time data analysis, which could challenge deployment on devices with limited processing capabilities. In contrast, KFPSO [33] combines the Kalman filter and particle swarm optimization (PSO) to dynamically adjust the filter parameters. This hybrid approach aims to improve the accuracy of state estimation and anomaly detection by improving real-time noise variations. Despite its effectiveness, KFPSO introduces significant computational complexity due to the optimization process, which makes it less suitable for WSNs with limited resources.

The discussion emphasizes exploring different anomaly detection methodologies in WSNs, highlighting the balance between cost and efficiency [34]. It showcases the potential of the Kalman filter as a fast and lightweight model despite performance limitations amid sensor fluctuations. Therefore, our proposed OAKF framework offers several innovative contributions to address the existing challenges in WSNs anomaly detection:Unlike traditional Kalman filters with fixed parameters, OAKF dynamically adjusts for noise variations in the process and measurement in real time. This ensures a high filtering performance even under different environmental conditions and sensor noise characteristics.The OAKF framework is specifically designed for real-time applications. It can quickly adapt to changing data streams, ensuring accurate and timely anomaly detection without significant computational costs.The OAKF algorithm is optimized for use in resource-constrained sensor nodes, making it highly scalable and computationally efficient. It can be deployed in large-scale wireless sensor networks without compromising the detection accuracy or response time.

## 3. Proposed Model

The goal of the proposed Online AKF (OAKF) framework is to detect anomalies in real-time data streams from spatially distributed sensors. The streams are divided into equal-sized intervals containing N continuous samples (sequence of sensor readings) for each sensor, as illustrated in Figure 1. For this task, we relied on a dataset released by the Intel Berkeley Research Laboratory, which included temperature measurements from 54 device sensors at one-minute time intervals [35].

Based on Figure 1, the OAKF framework operates by adapting two variables, *R* (measurement noise covariance) and *Q* (process noise covariance), to achieve high-performance anomaly detection without a significant increase in the computational cost. In this work, we define an anomaly as a large deviation from the expected state [36], where the innovation exceeds the maximum threshold, indicating a possible sensor anomaly. Moreover, the noise in the sensor measurements is modeled as Gaussian noise [37], assuming that both the *R* and the *Q* are Gaussian white noise with a zero mean. 

In OAKF, higher innovation values result in gradual increases in the *Q* and *R*, which helps the filter adapt to changing noise levels. Higher noise power increases uncertainty, which can lead to more false positives and negatives. Conversely, lower noise power results in more reliable measurements and stable operation, which enhances accuracy by reducing false alarms. OAKF dynamically adjusts *Q* and *R* based on innovations in real time, maintaining a high filtering performance and robust anomaly detection despite changing noise characteristics. This adaptability ensures the effective management of noise levels, improving the overall detection accuracy, as we discuss below. For further clarification, Table 1 summarizes the notation list used in this proposal.

However, KF operates in two main steps: prediction and update. In the prediction step, the Predicated State Estimate is illustrated as shown in Equation (1):(1)$${\hat{x}}_{k|k-1}={\hat{x}}_{k-1|k-1}$$

Here, x represents the sensor measurement, and *k* represents the time step. This equation assumes a simple model where the next state is equal to the current estimate. The Predicated State Covariance is updated using: (2)$$P_{k|k-1}=P_{k-1|k-1}+Q$$

In this equation, P is the estimate covariance, and Q is the process noise covariance. This step updates the estimate’s uncertainty by adding the process noise, reflecting the system’s inherent unpredictability.

During the update steps, the Kalman Gain (K) is calculated to determine the weight given to the new measurement versus the prediction, as illustrated in Equation (3).
(3)$$K_{k}=\frac{P_{k|k-1}}{P_{k|k-1}+R}$$

Here, R is the measurement noise covariance. The Updated State Estimate is computed as follows: (4)$${\hat{x}}_{k|k}={\hat{x}}_{k|k-1}+K_{k}(z_{k}-{\hat{x}}_{k|k-1})$$
where $z_{k}$ is the actual measurement. This equation corrects the predicted state using the measurement and the Kalman Gain. Finally, the Updated Estimate Covariance is calculated to reduce uncertainty, as shown in Equation (5).
(5)$$P_{k|k}=(1-K_{k})P_{k|k-1}$$

To adaptively adjust the parameters (Q and R), the OAKF framework uses the innovation sequence, which is the difference between the actual measurement and the predicted state $\left(z_{k}-{\hat{x}}_{k|k-1}\right)$. The adjustments are made based on the following conditions:(6)$$if\;\left(z_{k}-{\hat{x}}_{k|k-1}\right)>Threshold\;\left\{\begin{array}{c} Q=\min(Q+\Delta Q,\;Q_{max})\\ R=\min(R+\Delta R,\;R_{max}) \end{array}\right.$$

This adaptive mechanism ensures that the filtering parameters are dynamically adjusted in response to live data, enhancing the filter’s performance in detecting anomalies amidst sensor noise and environmental changes. The ΔQ and ΔR are small increments, and $Q_{max}$ and $R_{max}$ are the maximum allowed values for Q and R, respectively.

The OAKF technique for real-time anomaly detection in sensor measurements can be outlined as shown with Algorithm 1.
**Algorithm 1: OAKF**1. Start2. Set x^0|0 to the first sensor measurement or a known initial condition3. Initialize P0|0, Q, and R based on prior knowledge or estimation4. Define thresholds for anomaly detection and adaptive adjustment (ΔQ, ΔR, Qmax, Rmax)5. Read sensor data periodically (zk) 6. **While**
zk
**do**
7. **Prediction**:7.1 Predicated State Estimate (zk)7.2 Update State Covariance (zk)8. **Update**:8.1 Find Kk8.2 Update the state estimate with the new measurement (x^k|k)8.3 Update the estimate covariance (Pk|k)9. **Adaptive Adjustment**:9.1 Calculate the innovation (innovation=zk−x^k|k−1)9.2 Adjust Q and R based on Equation (6) 10. **Anomaly Detection**: 10.1 **for** each k10.2  if innovation>predefined_anomaly_threshold10.3   Flage (zk) = −110.4  **else if**
10.5   Flage (zk) = 110.6  **end if**10.7 **end for**11. **end while**12. threshold = std(window_innovations) * 2 // set threshold as twice the moving standard deviation13. **End**

## 4. Results and Discussion 

In this section, we will delve into the aspects of dataset collection, the incidence of deviations, and the corresponding remediation procedures.

### 4.1. Dataset Collection 

Our study rigorously tested a new methodology using real-time data from WSNs deployed in the IBRL [35]. We curated a dataset from 54 sensors, focusing on temperature measurements at one-minute intervals and a bandwidth of 12.4 kbps. The initial dataset comprises approximately 2.5 million records and includes attributes such as the date, time, node ID, epochs, temperature, humidity, light, voltage, and location of the nodes. Data cleansing was vital to remove inconsistencies, narrowing down from eight attributes to only temperature. This refinement involved filtering out anomalies and irregularities, such as instances where the temperature value reached 120 degrees. Moreover, we conducted a visual analysis by plotting the first 2500 samples of test data from sensor 4 for in-depth analysis, as presented in Figure 2. 

### 4.2. Incidence of Deviations

To assess our method’s impact on detecting anomalies, we introduced controlled offsets (Υ) to the data from a low-cost sensor (node no. 4), mimicking imprecision. These offsets, ranging from −3 to +3, were uniformly added to the node’s readings. The offsets were applied as described by Equation (7):(7)$$Un=\left\{\begin{array}{c} v_{i}+\Upsilon ,\;\;\;\;\;\;\;\;0<\Upsilon <3\;\;\;\;\\ v_{i}+\Upsilon ,\;\;-3\;\leq \Upsilon <0\; \end{array}\right.$$

Figure 3 illustrates the results after applying Equation (7) to sequences of 40 to 60 readings from node 4. 

### 4.3. Detection Procedure

In this section, we explore the impact of drift detection on the accuracy of WSN node readings. We assess the efficacy of our methodology through two key metrics: accuracy and time efficiency. Our strategy leverages unsupervised learning techniques. For accuracy measurement, labels are assigned to original readings devoid of any offsets, while readings with errors are marked to facilitate accuracy evaluation using a confusion matrix approach. The formula for accuracy is:(8)$$Accuracy=\frac{True\;Positives+True\;Negatives}{True\;Positives+True\;Negatives+False\;Positives+False\;Negatives}\;$$

For time consumption, the equation can be:(9)Total Processing Time=total_end_time−total_start_time

Our analysis, conducted through Python simulations on a machine with a 1.8 GHz Core i5 processor, 8 GB cache, and 12 GB RAM [11], also includes the initial parameter values detailed in Table 2.

In the context of our analysis, it is critical to acknowledge the deliberate calibration of key parameters to refine our outcomes. This meticulous process involved adjusting variables such as the initial estimate covariance, the process noise covariance (*Q*), the measurement noise covariance (*R*), the threshold for detecting anomalies, and the window size for analyzing innovations. The chosen values are as follows: an initial estimate covariance of 1 × 10^−4^, a process noise covariance (*Q*) set at 1 × 10^−5^, a measurement noise covariance (*R*) also at 1 × 10^−5^, a threshold for detecting anomalies set at 2.0, and a window size for innovations set to 500. These values were chosen based on experimental tests, ensuring a balance between the sensitivity to distortions and the stability of the filter. As a result, these adjustments were useful in improving the performance of the system, ensuring more accurate results for anomaly detection and state estimation. Moreover, it is important to note that the values chosen for the optimization parameters are specific to the dataset used in this study. 

Additionally, the false positive rate (FPR) was employed to assess the accuracy calculation, as depicted in Figure 4. Real sensor readings were assigned a label of 1, while distorted readings were labeled as −1. Despite the inherent fluctuations and abrupt changes in the actual sensor readings (e.g., at 500 and 1940), the proposed OAKF model achieved an impressive accuracy (based on FPR) of 99.6% in aligning sensor readings (labeled) with predictions generated during the training process. The color red represents the FPR, while the color blue signifies the matching values between OAKF predictions and the actual sensor readings. 

The reason for this accuracy is due to the algorithm’s ability to calibrate the R and *Q*. The filter adapts its process noise covariance (*Q*) and measurement noise covariance (*R*) based on the magnitude of the innovations (the difference between the measured temperature and the estimated state from the previous time step). Moreover, surprisingly, the computational cost of these processes did not exceed 0.008 s, since the STD calculation is performed after every 500 readings. On the other hand, there was no adjustment to the threshold because the level of data volatility was not outside the range of the initial value. 

In the process of generating the displacement and implementing OAKF on those decisions, as shown in Figure 5, OAKF gave excellent outputs, with the accuracy exceeding approximately %94.27 and the execution time approximately 0.0080 s. The accuracy of the results is due to the ability to adapt the *Q* and *R*.

The figure also shows the ability to adapt the threshold to changing readings. Moreover, the reason it costs so little is that the threshold adaptation is not applied continuously. 

In analyzing the effect of the data size on the OAKF model’s accuracy and execution time, we determined the relationship, as shown in Figure 6.

The figure shows a decline in the accuracy from 95.4% for the dataset 1250 to 94.3% for the dataset 2500, while in the rest of the datasets, the accuracy is close. The reason for this is that the largest part of the readings in the first dataset does not contain a drift; until reading 700, there is no drift. After 2500 readings, the accuracy variance starts to decrease with the increasing dataset, and the explanation for the observed decline in the accuracy in the initial part of the dataset and the subsequent stability is that the first part and up to 4000 readings represent the stage of adaptation to the values of the process noise covariance (*Q*) and measurement noise covariance (*R*). After this adaptation phase, the stability of the accuracy is due to the continuous adjustment of *Q* and *R*, which allows the OAKF framework to maintain a high filtering performance even as the dataset size increases. Moreover, the time cost increases regularly due to the size of the dataset. The cost increased from 0.001 s in the smallest dataset (1250) to 0.0197 s in the fourth dataset (8000). This time is considered very short compared to other methods, as we will see later.

By comparing the proposed model (OAKF) with other methods, we selected different techniques and recent studies. KFPSO [33], standard AKF [22], the Kalam AE [28], DWR-K-means [6] were used in this analysis, as illustrated in Figure 7. 

The OAKF model showcases a high performance in the domain of anomaly detection across datasets of varying sizes, indicating its robust adaptability and efficiency. Unlike standard Kalman filters, OAKF utilizes a dynamic adjustment mechanism for its process and measurement noise covariances (denoted as *Q* and *R* in the code), which are fine-tuned based on the discrepancies between predicted and actual measurements, termed innovations. The OAKF framework’s enhanced ability to adjust its parameters in real time to the evolving statistical properties of the dataset likely contributes to its consistently high accuracy, as it can better handle non-linear patterns and subtle anomalies that may be present in larger datasets. This adaptability is crucial for maintaining precision in state estimation and anomaly detection in complex systems, where data variability is common. Furthermore, the ability to adapt threshold_anomlay detection helped achieve this accuracy. 

In addition, the Kalam AE also gave a high accuracy; however, OAKF’s approach may lead to faster and more responsive adjustments to the observed data, which could explain its higher accuracy in anomaly detection. The deep embedding optimization in the Kalman AE, while powerful, might be more suited to capturing complex patterns rather than quick adaptation, which can be crucial depending on the nature of the dataset and the anomalies present. Furthermore, OAKF might outperform KFPSO if its adaptation mechanism more closely aligns with the actual changes in the data, providing more accurate estimates and anomaly detection. Conversely, if the PSO in KFPSO is able to find a set of parameters that are near optimal and the data does not change too dramatically over time, KFPSO could also show a strong performance. The observed higher accuracy of OAKF suggests that for the given datasets, its adaptive mechanism may be more aligned with the data’s characteristics, leading to better performance compared to the potentially static optimization of KFPSO. Also, the advanced accuracy of OAKF over DWT-K-means for anomaly detection in time series data can be attributed to its real-time adaptability, dynamic parameter optimization, and potentially lower computational complexity. While DWT-K-means provides a robust method for feature extraction and the identification of anomalies at different scales, its static nature and computational demands may limit its effectiveness and efficiency compared to the more adaptive and streamlined approach of OAKF.

OAKF exhibits a notable improvement in accuracy over other methods, with gains of 2.325% over AKF, 1.245% over the Kalman AE, a significant 8.278% over DWT-K-means, and 1.483% over KFPSO, underscoring its effectiveness in anomaly detection across varying dataset sizes.

The process cost is very important in wireless sensor networks because the sensors work with a microprocessor, as we explained previously. Figure 8 presents a clear trade-off between the computational time and complexity of the methods. 

Based on the provided time consumption comparison figure, AKF appears to have the lowest time consumption, suggesting efficient computation particularly with smaller datasets. As the dataset size increases, OAKF maintains a competitive time efficiency, outperforming the more computationally intensive methods such as DWT-K-means and KFPSO. This indicates that while OAKF achieves a high accuracy, it does not do so at the cost of excessive computational demands, striking a balance between performance and efficiency. However, the Kalman AE is not included in Figure 8 because it required about 39.16 s during the training process.

The average time consumption data reveals that OAKF is more time consuming compared to the KF, with OAKF taking approximately 700% more time on average. Despite this, OAKF is significantly more time efficient than both DWT-K-means and KFPSO, which take approximately 7737.9% and 24,583.5% more time than OAKF, respectively. This suggests that while OAKF does not have the lowest time consumption, it strikes a balance by outperforming more complex methods that take considerably longer to execute, offering a middle ground in terms of computational efficiency. 

In a summary of the comparative analysis, Table 3 shows the comparisons between anomaly detection techniques.

The proposed OAKF framework demonstrated a balance between high accuracy (95.4%) and low false positive rate (3.0%), with a minimum processing time of 0.008 s per sample. This efficiency is achieved by dynamically adjusting filtration parameters based on the innovation sequence, ensuring robustness against varying noise characteristics and environmental changes. Unlike traditional methods that rely on fixed parameters or complex optimization processes, OAKF constantly adapts to evolving statistical properties of the data.

## 5. Conclusions and Future Work

The work presented effectively addresses the pivotal challenge of real-time anomaly detection in WSNs by proposing a new framework, called Online Adaptive Kalman Filtering (OAKF). OAKF stands out due to its dynamic parameter adjustment capabilities, enabling it to adeptly handle the intrinsic noise and variability of low-cost sensor data along with adaptive anomaly detection_Threshold. This innovation not only advances the accuracy of anomaly detection but also does so while preserving the computational efficiency, a critical aspect for the deployment in resource-constrained sensor nodes. The OAKF framework demonstrated a balance between high accuracy (95.4%) and low FPR (3.0%), with a minimum processing time of 0.008 s per sample.

The comparative analysis demonstrates OAKF’s better performance in terms of accuracy against established methods like AKF, the Kalman AE, DWT-K-means, and KFPSO. Notably, it offers significant improvements, especially over DWT-K-means, while maintaining a competitive time efficiency even as dataset sizes increase. This is particularly important given the limited computational capabilities of typical WSN nodes. AKF showed a significant improvement in accuracy over other methods, with gains of 2.325% over AKF, 1.245% over the Kalman AE, 8.278% over DWT-K-means, and 1.483% over KFPSO. Moreover, OAKF is significantly more time efficient than both DWT-K-means and KFPSO, which take about 7737.9% and 24,583.5% longer than OAKF, respectively. 

In future work, we need to conduct an analytical study on the impact of OAFK on different types and datasets.

## Figures and Tables

**Figure 1 sensors-24-05046-f001:**
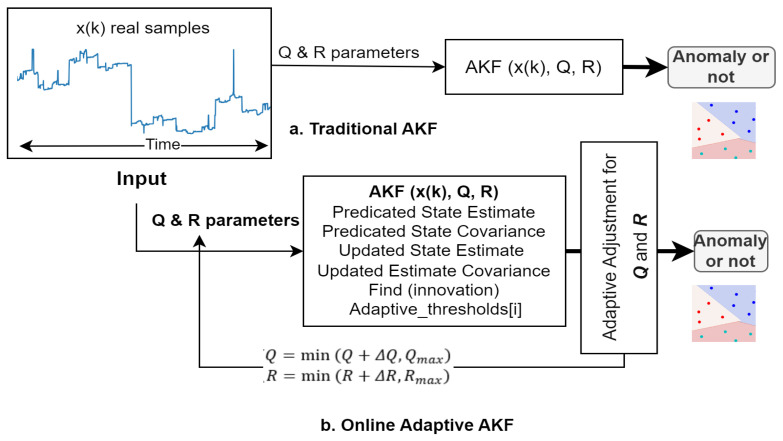
The difference between the traditional AKF and the proposed strengthening of the AKF.

**Figure 2 sensors-24-05046-f002:**
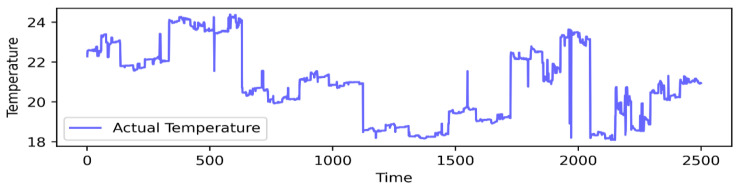
Sample of sensor 4 reading measurements.

**Figure 3 sensors-24-05046-f003:**
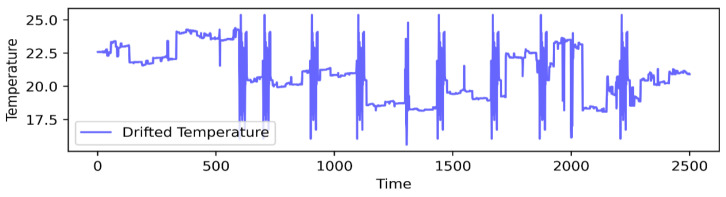
Sample of sensor 4 drifted measurements.

**Figure 4 sensors-24-05046-f004:**
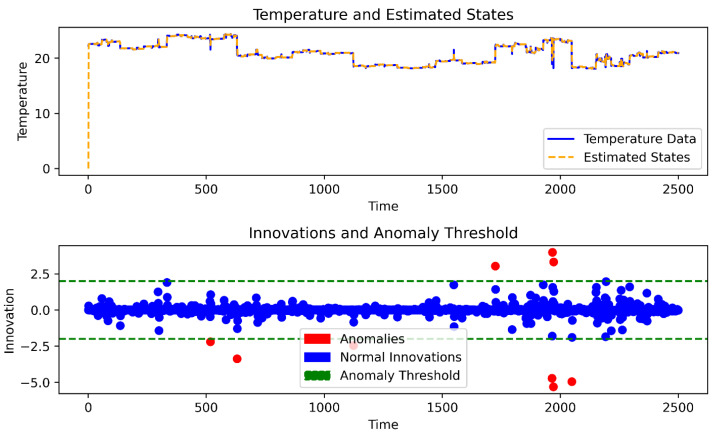
FPR (anomalies) post-OAKF.

**Figure 5 sensors-24-05046-f005:**
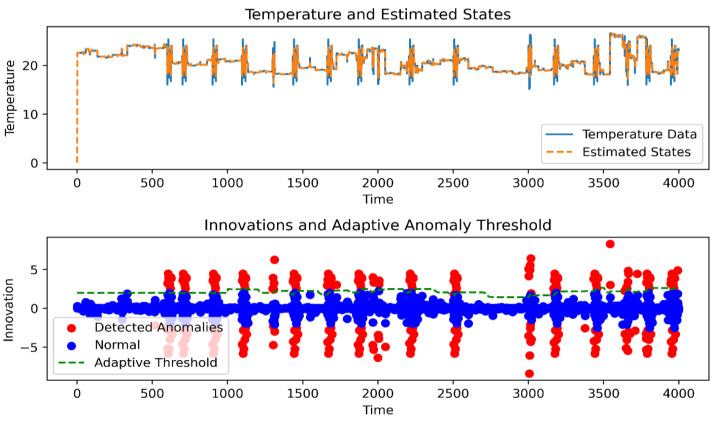
OAKF anomaly for 4000 drifted readings.

**Figure 6 sensors-24-05046-f006:**
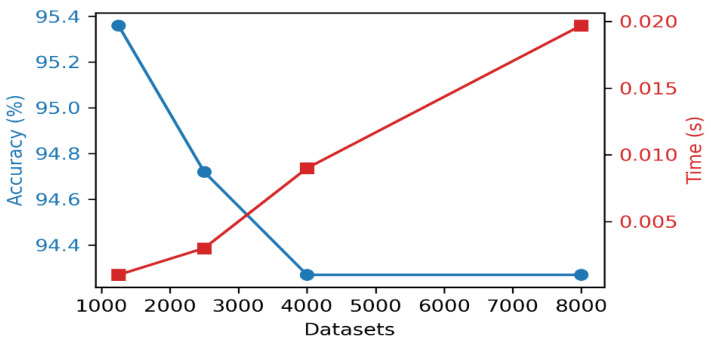
OAKF’s accuracy and execution time vs. dataset size.

**Figure 7 sensors-24-05046-f007:**
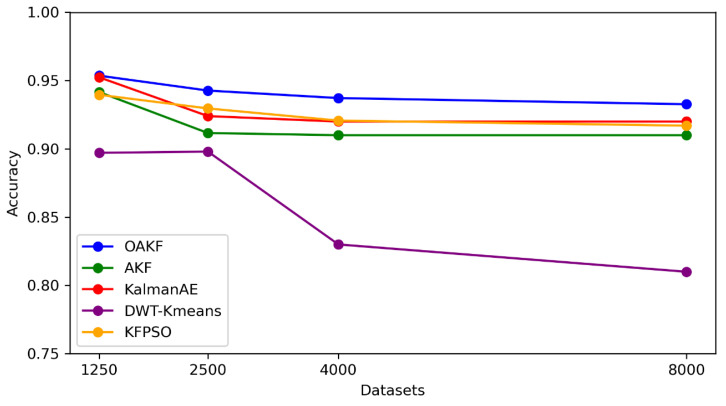
Accuracies of different methods.

**Figure 8 sensors-24-05046-f008:**
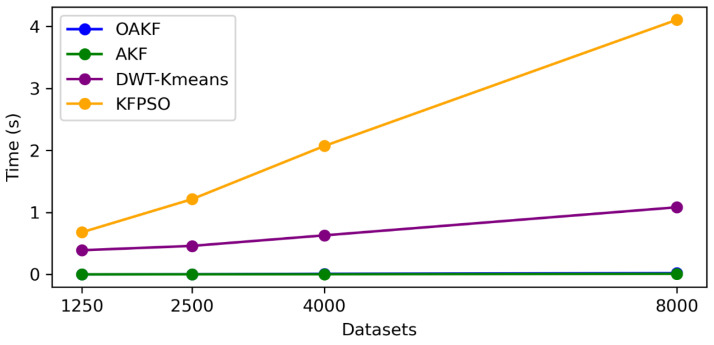
Time costs for different methods.

**Table 1 sensors-24-05046-t001:** Notation list.

Symbol	Description
x	Sensor measurement
k	Time step
P	Estimate covariance
Q	Process noise covariance
K	Kalman Gain
z	Actual measurement
ΔQ	Increment for process noise covariance
ΔR	Increment for measurement noise covariance
$Q_{max}$	Maximum value for measurement noise covariance
innovation	Difference anomaly detection threshold
threshold	Predefined anomaly detection threshold

**Table 2 sensors-24-05046-t002:** Initial parameter values of OAKF.

Variable	Value
initial_estimate_covariance	1 × 10^−4^
initial_*Q*	1 × 10^−5^
initial_*R*	1 × 10^−5^
Initial_threshold_ detect_anomalies	2.0

**Table 3 sensors-24-05046-t003:** Comparative analysis of anomaly detection techniques.

Symbol	Accuracy (%)	FPR (%)	Time Cost (s)
Traditional KF [22]	91.0	22.0	0.001
KFPSO [33]	92.0	15.0	2.0
DWT-K-means [6]	85.0	30.0	0.6
Kalman AE [28]	95.2	4.0	39.1 (training)
OAKF (proposed)	95.4	3.0	0.008

## Data Availability

Not applicable, Existing data from IBRL Lab were used.

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
