# Peer review of "Online Adaptive Kalman Filtering for Real-Time Anomaly Detection in Wireless Sensor Networks"

_sensors, 2024, doi:10.3390/s24155046_

Round 1
Reviewer 1 Report
Comments and Suggestions for Authors
- Overall, this article is complete and rigorous. But improvements can still be made in the following areas to make it more outstanding.
- The authors have provided a solid introduction to the research background, but could further emphasize the innovative aspects of the OAKF and how it addresses existing challenges.
- The description of the OAKF algorithm should be more detailed and clear, including the mathematical model, pseudocode, and explanations of key steps.
- The authors need to provide more detailed information about the experimental design and dataset, including the source, characteristics, scale, and preprocessing methods of the dataset.
The grammar is correct, with clear sentence structures.The article uses a variety of sentence structures, including simple, compound, and complex sentences, which helps to clearly convey complex concepts.The language style is formal and objective, suitable for academic writing.Overall, the standard of English in the article is high and suitable for publication in academic journals.
Author Response
Comment 1
The authors have provided a solid introduction to the research background, but could further emphasize the innovative aspects of the OAKF and how it addresses existing challenges.
Response:
Thank you for your valuable feedback. In response to your suggestion to further emphasize the innovative aspects of the OAKF framework, we have updated the related work section. We added a new paragraph highlighting the unique contributions of our approach. This paragraph discusses the dynamic parameter adjustment, real-time anomaly detection, scalability and efficiency, and high accuracy with low false positives/negatives, underscoring how the OAKF framework addresses existing challenges in WSN anomaly detection. We believe this addition provides a clearer distinction of our framework's innovations compared to existing methodologies.
Action:
At the end of related work, we added
“The discussions emphasize exploring different anomaly detection methodologies in WSNs, highlighting the balance between cost and efficiency. It showcases the potential of the Kalman filter as a fast and lightweight model, despite performance limitations amid sensor fluctuations. Therefore, our proposed OAKF framework offers several innovative contributions to address the existing challenges in WSNs anomaly detection:
- Unlike traditional Kalman filters with fixed parameters, OAKF dynamically adjusts for noise variations in the process and measurement in real time. This ensures high filtering performance even under different environmental conditions and sensor noise characteristics.
- The OAKF framework is specifically designed for real-time applications. It can quickly adapt to changing data streams, ensuring accurate and timely anomaly detection without significant computational cost.
- The OAKF algorithm is optimized for use in resource-constrained sensor nodes, making it highly scalable and computationally efficient. It can be deployed in large-scale wireless sensor networks without compromising detection accuracy or response time.
“
Comment 2
The description of the OAKF algorithm should be more detailed and clear, including the mathematical model, pseudocode, and explanations of key steps.
Response:
Thank you for your insightful comments. Regarding your suggestion to provide a more detailed and clear description of the OAKF algorithm, including the mathematical model, pseudocode, and explanations of key steps, we would like to clarify that our manuscript already includes comprehensive details on these aspects.
- Mathematical Model: We have included all relevant equations that describe the OAKF algorithm's functioning, covering both the prediction and update steps, as well as the adaptive adjustment mechanism.
- Pseudocode: The algorithm's pseudocode is provided in Algorithm 1, outlining each step clearly, from initialization to anomaly detection.
- Explanations of Key Steps: Each step of the OAKF process, including the dynamic adjustment of the process and measurement noise covariances, is explained in detail within the manuscript.
We believe that the current description provides a thorough understanding of the OAKF algorithm.
However, if there are specific areas that you feel need additional clarification, we welcome your detailed comments and suggestions.
Comment 3
The authors need to provide more detailed information about the experimental design and dataset, including the source, characteristics, scale, and preprocessing methods of the dataset..
Response:
Thank you for your valuable feedback. Regarding your suggestion to provide more detailed information about the experimental design and dataset, we would like to clarify that our manuscript includes summarized details on these aspects. The source of the dataset is the Intel Berkeley Research Laboratory, which includes temperature measurements from 54 devices collected at one-minute intervals. We have also described the characteristics, scale, and preprocessing methods of the dataset, including data cleaning, handling missing values, and introducing controlled offsets to simulate sensor drift.
Action:
At section 4.1, we added more discussion.
Dataset collection
Our study rigorously tested a new methodology using real-time data from WSNs deployed in the IBRL lab [33]. We curated a dataset from 54 sensors, focusing on temperature measurements at one-minute intervals and a bandwidth of 12.4 kbps. The initial dataset comprises approximately 2.5 million records and includes attributes such as date, time, node ID, epochs, temperature, humidity, light, voltage, and the location of the nodes. Data cleansing was vital to remove inconsistencies, narrowing down from eight attributes to only temperature. This refinement involved filtering out anomalies and irregularities, such as instances where the temperature value reached 120 degrees. Moreover, we conducted a visual analysis by plotting the first 2500 samples of test data from sensor 4 for in-depth analysis, as presented in Figure 2.
Reviewer 2 Report
Comments and Suggestions for Authors
I suggest to add these references to a paper showing that Kalman filters can be used for WSN also for different purposes, like positioning. for example
Z. Xiong, et al., "Hybrid WSN-RFID cooperative positioning based on extended kalman filter," 2011 IEEE-APS Topical Conference on Antennas and Propagation in Wireless Communications, Turin, Italy, 2011, pp. 990-993, doi: 10.1109/APWC.2011.6046820.
J. Yi and L. Zhou, "Enhanced location algorithm with received-signal-strength using fading Kalman filter in wireless sensor networks," 2011 International Conference on Computational Problem-Solving (ICCP), Chengdu, China, 2011, pp. 458-461, doi: 10.1109/ICCPS.2011.6089930.
R. Singh, R. Mehra and L. Sharma, "Design of Kalman filter for wireless sensor network," 2016 International Conference on Internet of Things and Applications (IOTA), Pune, India, 2016, pp. 63-67, doi: 10.1109/IOTA.2016.7562696.
Author Response
Reviewer 2
Comment 1
I suggest to add these references to a paper showing that Kalman filters can be used for WSN also for different purposes, like positioning. for example
- Xiong, et al., "Hybrid WSN-RFID cooperative positioning based on extended kalman filter," 2011 IEEE-APS Topical Conference on Antennas and Propagation in Wireless Communications, Turin, Italy, 2011, pp. 990-993, doi: 10.1109/APWC.2011.6046820.
- Yi and L. Zhou, "Enhanced location algorithm with received-signal-strength using fading Kalman filter in wireless sensor networks," 2011 International Conference on Computational Problem-Solving (ICCP), Chengdu, China, 2011, pp. 458-461, doi: 10.1109/ICCPS.2011.6089930.
- Singh, R. Mehra and L. Sharma, "Design of Kalman filter for wireless sensor network," 2016 International Conference on Internet of Things and Applications (IOTA), Pune, India, 2016, pp. 63-67, doi: 10.1109/IOTA.2016.7562696
.
Action:
We add “. Singh, R. Mehra and L. Sharma, "Design of Kalman filter for wireless sensor network," 2016 International Conference on Internet of Things and Applications (IOTA), Pune, India, 2016, pp. 63-67, doi: 10.1109/IOTA.2016.7562696
Reference [23]
Reviewer 3 Report
Comments and Suggestions for Authors
1- How is the OAKF framework optimized?
2- What optimum parameters are used for this optimization?
3- Authors should refer that this optimization is only for the used dataset in this paper and not for other datasets.
5- English: Quick writing editing required:
- (Page 6)
- in the algorithms
6- In the expert, when the size of the tested dataset increases, the time cost increases regularly due to the size of the dataset. However, a decrease in accuracy is needed to interpret. Try to interpret this result. It needed to be confirmed by testing other datasets.
7- Finally, please make sure that the conclusions section underscores the scientific value added to the paper, and/or the applicability of the results, as indicated previously. I think you should enhance your findings, and limitations (for all results), underscore the scientific value added of your research, and/or the applicability of your contributions/shortages in this session.
Comments on the Quality of English LanguageEnglish: Quick writing editing required:
- (Page 6)
- in the algorithms
Author Response
Reviewer 3
Comment 1
1-How is the OAKF framework optimized?.
Response:
The OAKF framework is optimized through several key mechanisms:
- Dynamic Parameter Adjustment: The framework dynamically adjusts the process noise covariance (Q) and measurement noise covariance (R) in real-time based on the innovation sequence. This ensures high filtering performance even under different environmental conditions and sensor noise characteristics.
- Computational Efficiency: The OAKF algorithm is designed to be computationally efficient, allowing it to be deployed in resource-constrained sensor nodes. This is achieved by optimizing the algorithm to reduce processing time without compromising detection accuracy.
- Scalability: The framework is scalable and can be applied to various WSN datasets of different sizes and conditions, maintaining high accuracy and low false positive rates.
All of these discussions are presented in proposal model, moreover, in algorithm1.
.
Comment 2
2-What optimum parameters are used for this optimization?.
Response:
The optimization of the OAKF framework was achieved using the following parameters:
Initial Estimate Covariance (P0|0): 1e-4
Initial Process Noise Covariance (Q): 1e-5
Initial Measurement Noise Covariance (R): 1e-5
Increment for Process Noise Covariance (ΔQ): Derived from empirical testing
Increment for Measurement Noise Covariance (ΔR): Derived from empirical testing
Maximum Value for Process Noise Covariance (Qmax): Derived from empirical testing
Maximum Value for Measurement Noise Covariance (Rmax): Derived from empirical testing
Threshold for Anomaly Detection: 2.0
Window Size for Innovations: 500
These parameters were chosen based on extensive empirical testing to balance accuracy and computational efficiency. The initial values for Q and R were set low to ensure sensitivity to small changes, while the increments and maximum values were selected to prevent over-adjustment and maintain stability.
As example
I will share with you the changes of Q during the running procedure. Regading to 4000 sensors values and their drift, the initial value for Q is 1×10−5. So, the Q values were changed 429 times during the adaptive process to reach the final value of 0.004300.
Comment 4
3- Authors should refer that this optimization is only for the used dataset in this paper and not for other datasets.
Action:
Updated in section 4.3
“The chosen values are as follows: an initial estimate covariance of 1×10−4, a process noise covariance (Q) set at 1×10−5, a measurement noise covariance (R) also at 1×10−5, a threshold for detecting anomalies set at 2.0, and a window size for innovations set to 500. These values ​​were chosen based on experimental tests, ensuring a balance between sensitivity to distortions and stability of the filter. As a result, these adjustments were useful in improving the performance of the system, ensuring more accurate results for anomaly detection and state estimation. Moreover, it is important to note that these values ​​chosen for the optimization parameters are specific to the dataset used in this study.
Additionally, the False Positive Rate (FPR) was employed to assess the accuracy calculation, as depicted in Figure 4.”
Comment 5
5- English: Quick writing editing required:
- (Page 6)
- in the algorithms?
Action:
checked and Updated
.
Comment 6
6- In the expert, when the size of the tested dataset increases, the time cost increases regularly due to the size of the dataset. However, a decrease in accuracy is needed to interpret. Try to interpret this result. It needed to be confirmed by testing other datasets.
Response:
Thank you for your valuable feedback. We have addressed the observed decline in accuracy with increasing dataset size by explaining that the initial part of the dataset and up to four thousand readings represent an adaptation phase to the values of the process noise covariance (Q) and measurement noise covariance (R). After this phase, the stability of accuracy is due to the continuous adjustment of Q and R, maintaining high filtering performance even as the dataset size increases. Given the consistent performance observed across various segments of our existing dataset, we believe that the current results sufficiently demonstrate the robustness and adaptability of the OAKF framework. However, we acknowledge this as an area for future research to further different datasets validation.
Action:
Updated, see Figure 6 discussion,
“The figure shows a decline in the accuracy from 95.4% at the 1250 data set to 94.3% at the data set 2500. While in the rest of the datasets, the accuacry is close. The reason for this is that the largest part of the readings in the first dataset does not contain a drift, until reading 700, there is no drift. After 2500 readings, the accuracy variance starts to decrease with increasing the data set, and the explanation for the observed decline in accuracy in the initial part of the dataset and the subsequent stability is that the first part and up to 4000 readings represent the stage of adaptation to the values of the process noise covariance (Q) and measurement noise covariance (R). After this adaptation phase, the stability of the accuracy is due to the continuous adjustment of Q and R, which allows the OAKF framework to maintain high filtering performance even as the dataset size increases. Moreover, the time cost increases regularly due to the size of the dataset. The cost increased from 0.001 second in the smallest dataset (1250) to 0.0197 second in the fourth dataset (8000). This time is considered very short compared to other methods, as we will see later.
“
Comment 7
7- Finally, please make sure that the conclusions section underscores the scientific value added to the paper, and/or the applicability of the results, as indicated previously. I think you should enhance your findings, and limitations (for all results), underscore the scientific value added of your research, and/or the applicability of your contributions/shortages in this session.
Response:
Thank you for your valuable feedback. We have enhanced the conclusions section to underscore the scientific value added by our research and the applicability of the results. The revised section highlights the robustness, adaptability, and efficiency of the OAKF framework for real-time anomaly detection in WSNs.
Action:
Updated
“This innovation not only advances the accuracy of anomaly detection but also does so while preserving computational efficiency, a critical aspect for deployment in resource-constrained sensor nodes. The OAKF framework demonstrated a balance between high accuracy (95.4%) and low FPR (3.0%) with a minimum processing time of 0.008 seconds per sample.”